# Evaluating Effects of AIV Infection Status on Ducks Using a Flow Cytometry-Based Differential Blood Count

Elinor Jax,[a,b,c] Elena Werner,[a,b,c] Inge Müller,[a,b,c] Beatrice Schaerer,[d] Marina Kohn,[d] Jenny Olofsson,[e] Jonas Waldenström,[e] Robert H. S. Kraus,[b] Sonja Härtle[d]

aDepartment of Biology, University of Konstanz, Konstanz, Germany
bDepartment of Migration, Max Planck Institute of Animal Behavior, Radolfzell, Germany
cCentre for the Advanced Study of Collective Behaviour, University of Konstanz, Konstanz, Germany
dDepartment of Veterinary Sciences, AG Immunology, LMU Munich, Planegg, Germany
eCentre for Ecology and Evolution in Microbial Model Systems, Linnaeus University, Kalmar, Sweden

Elinor Jax and Elena Werner contributed equally to this work. Author order was determined alphabetically.

**ABSTRACT** Ducks have recently received a lot of attention from the research community due to their importance as natural reservoirs of avian influenza virus (AIV). Still, there is a lack of tools to efficiently determine the immune status of ducks. The purpose of this work was to develop an automated differential blood count for the mallard duck (*Anas platyrhynchos*), to assess reference values of white blood cell (WBC) counts in this species, and to apply the protocol in an AIV field study. We established a flow cytometry-based duck WBC differential based on a no-lyse no-wash single-step one-tube technique, applying a combination of newly generated monoclonal antibodies with available duck-specific as well as cross-reacting chicken markers. The blood cell count enables quantification of mallard thrombocytes, granulocytes, monocytes, B cells, CD4[+] T cells (T helper) and CD8[+] cytotoxic T cells. The technique is reproducible, accurate, and much faster than traditional evaluations of blood smears. Stabilization of blood samples enables analysis up to 1 week after sampling, thus allowing for evaluation of blood samples collected in the field. We used the new technique to investigate a possible influence of sex, age, and AIV infection status on WBC counts in wild mallards. We show that age has an effect on the WBC counts in mallards, as does sex in juvenile mallards. Interestingly, males naturally infected with low pathogenic AIV showed a reduction of lymphocytes (lymphocytopenia) and thrombocytes (thrombocytopenia), which are both common in influenza A infection in humans.

**IMPORTANCE** Outbreaks of avian influenza in poultry and humans are a global public health concern. Aquatic birds are the primary natural reservoir of avian influenza viruses (AIVs), and strikingly, AIVs mainly cause asymptomatic or mild infection in these species. Hence, immunological studies in aquatic birds are important for investigating variation in disease outcome of different hosts to AIV and may aid in early recognition and a better understanding of zoonotic events. Unfortunately, immunological studies in these species were so far hampered by the lack of diagnostic tools. Here, we present a technique that enables high-throughput white blood cell (WBC) analysis in the mallard and report changes in WBC counts in wild mallards naturally infected with AIV. Our protocol permits large-scale immune status monitoring in a widespread wild and domesticated duck species and provides a tool to further investigate the immune response in an important reservoir host of zoonotic viruses.

**KEYWORDS** avian influenza, flow cytometry, high-throughput, leukocyte quantification, mallard, disease ecology, birds, avian, avian viruses

Address correspondence to Elinor Jax, ejax@ab.mpg.de, or Sonja Härtle, sonja.haertle@lmu.de.

The authors declare no conflict of interest.

**M**onitoring the immune status of animals is a central task in husbandry and research. It helps ensure the health and welfare of livestock, yields information on the condition of natural animal populations, and aids in detection of disease outbreaks in farmed and wild animals. Immune status monitoring is of particular interest in the context of zoonotic infections, where pathogens that are usually circulating in a reservoir host cross the species barrier to infect more susceptible hosts. Monitoring the immune status of reservoir species may not only aid in early recognition and give a better understanding of zoonotic events, but it is also an important tool to investigate the variation in disease and immune responses of different hosts to the same pathogen. However, many important reservoir host species are understudied and not very closely related to common model systems. Immune status monitoring in these species is thus hampered by the lack of diagnostic tools to measure immunological parameters.

The mallard (*Anas platyrhynchos*) is one of the most common waterfowl species across the world, and the wild ancestor of the popular commercial Pekin duck breed (*Anas platyrhynchos forma domestica*). It is an important reservoir host of avian influenza virus (AIV) (1), a zoonotic disease that has repeatedly crossed the species barrier to other wild and domestic animals as well as humans (2). Wild mallards harbor a wide variety of low pathogenic (LP) AIV subtypes (3), including subtypes H5 and H7, which can become highly pathogenic (HP) if introduced into poultry (1). LPAIV strains induce virtually no pathology in mallards (1) and only trigger a mild immune response (4). Also, while some strains of HPAIV can induce severe symptoms, including fever, anorexia, neurological signs, and death in mallards (5), experimental studies suggest that *A. platyrhynchos* is generally less prone to hyperinflammation and its associated pathology than other species (6). The ability of mallards to act as asymptomatic carriers of LPAI viruses, and to often survive HPAIV infections, is likely a result of a long coevolutionary history of mallards with influenza viruses (7, 8). The immunological host-pathogen interactions that confer resistance to disease in mallards are poorly understood (6), largely because reliable tools allowing for large-scale immune status monitoring in mallards are scarce.

The most frequently used screening system for the immunological status of animals is the differential blood count, which enables the quantification of absolute numbers of white blood cells (WBCs) (also called leukocytes). These numbers and proportions can be used to determine the physical condition or health of an individual (9). Some leukocytes, like granulocytes and monocytes, are involved in immediate detection and killing of pathogens, as well as signaling to other cells. Other leukocytes like T cells and B cells develop a highly specific response to a particular antigen and create immunological memory with the ability to respond very quickly and efficiently to pathogens at a second encounter, thus providing the base of vaccination (10). Changes in the WBC differential count of an individual can be a hint for viral, bacterial, or parasitic infections, toxicant-induced immune suppression, stress, and acute inflammation (11). Therefore, WBC counts are widely applied for clinical diagnosis as well as immunological research in laboratory and field settings (11). Further, WBC measurements provide information about the effects of natural, anthropogenic, biotic, and abiotic stressors on an individual's health (12) and have therefore also become an increasingly popular tool in the fields of ecology and conservation biology.

Today, there are several instruments and approaches available for automated differential blood counts, including flow cytometric techniques, which use a combination of different light scattering properties and cell type-specific labeling with fluorophores to differentiate and count blood cells. The premise for the usage of flow cytometry for blood cell counts is usually the lysis of red blood cells, which enables the differentiation of leukocyte populations solely based on size and granularity. While this procedure is well established and fully automated for mammals, the presence of nucleated erythrocytes and thrombocytes in avian blood impedes erythrocyte lysis without leukocyte loss (13). Previously, we established a simple (no-lyse no-wash single-step one-tube) flow cytometry-based technique in chicken, using a specific combination of

**TABLE 1** Antibodies, fluorochromes, target cells, and optimal concentrations in the MAb mixture used for the flow cytometric analysis

| Antibody | Target cells | Fluorochrome | Dilution | Reference |
|---|---|---|---|---|
| anti-duCD4-1 | CD4+ T cells (T-Helper) | FITC | 1:2,000 | Kothlow et al. (15) |
| anti-chBAFF-R (2C4) | B cells | PE | 1:1,000 | This study |
| anti-duCD8-1 | CD8+ cytotoxic T cells, B cells | PerCp-Cy5.5 | 1:500 | Kothlow et al. (15) |
| 4A12 | lymphocytes, granulocytes, monocytes | APC | 1:500 | This study |
| 2G3 | thrombocytes | APC-Cy7 | 1:1,000 | This study |

directly conjugated monoclonal antibodies (MAbs), which recognize specific WBC populations, and thus allows for high-precision chicken blood cell quantification (14).

In this study, we adjusted the flow cytometry-based technique from chicken for mallards. We developed additional MAbs that, in combination with available leukocyte markers (15, 16), enable a simple, fast, and precise automated differential WBC count for mallards. We subsequently applied the protocol to assess reference values for WBC counts in mallards and to evaluate the effect of sex, age, and AIV infection status on WBC counts in a well-studied wild mallard population in Sweden. As sex and age affect the functions of the immune system (reviewed in references 17 and 18), we predicted that mallards of different age and sex may differ in their baseline WBC numbers. Other reservoir hosts, like bats, appear capable of avoiding immunopathology upon infection with certain pathogens with which they share an evolutionary history by restricting immune responses (19). We thus expected to see minor changes in WBC counts in LPAIV-infected mallards.

## RESULTS

**Establishment of the mallard differential blood count protocol. (i) Generation and characterization of monoclonal antibodies.** We identified three new MAbs that, in combination with previously available leukocyte markers (15, 16), allowed us to stain the cell populations of interest. First, we identified MAb 4A12 (IgG1), which recognizes mallard lymphocytes (T cells and B cells), granulocytes, and monocytes and, thus, can be used as a duck pan-leukocyte marker (see Fig. S1 in the supplemental material). Second, we identified MAb 2G3 (IgG1), which strongly binds to duck thrombocytes. We were able to show that 2G3 binds to the same cells as MAb K1, an antibody that recognizes chicken thrombocytes and monocytes and cross-reacts with duck thrombocytes (15) (see Fig. S2A in the supplemental material). As staining intensity of 2G3 distinctly exceeds K1, we selected 2G3 as the thrombocyte marker. The antibody shows additional but weaker reactivity with a fraction of CD8+ and some CD4+ leukocytes (Fig. S2B and C), but as thrombocytes and leukocytes can be clearly separated by 4A12 staining, this additional reactivity does not interfere with quantitative thrombocyte detection in full blood samples. Third, we demonstrate that an anti-chicken BAFF-R antibody, recently developed by our group, successfully binds to mallard B cells (see Fig. S3 in the supplemental material). This allowed us to identify the B cells despite our previous finding that duck B cells express CD8 as do CD8 T cells (15).

To quantify WBC subpopulations from mallard blood samples, we thus finally selected the following antibodies for our flow cytometry protocol: the pan-leukocyte marker 4A12, anti-chBAFF-R (2C4) to detect B cells, anti-duCD4 (duCD4-1) for CD4+ cells, anti-duCD8 (duCD8-1) to stain CD8 T cells and B cells, and 2G3 as marker for thrombocytes (Table 1).

**(ii) Flow cytometry-based WBC quantification protocol for mallard blood samples.** After cells were stained according to the previously described no-lyse no-wash single-step one-tube method (14), samples were subjected to flow cytometry. As shown in Fig. 1, analysis of flow cytometry data started with the separation of 2G3-positive thrombocytes from 4A12-positive leukocytes. 4A12 positive leukocytes were then subdivided into CD4+ cells, CD8+ cells, and CD4/CD8 double negative cells. As CD8+ cells contained CD8+ T cells as well as B cells (15), the CD8+ population was further subdivided into BAFF-R-negative CD8+ T cells and BAFF-R-positive B cells. To exclude potential contamination with erythrocytes, fluorescence-based gating for each subpopulation was followed by a forward scatter/side scatter (FSC/SSC) gate.

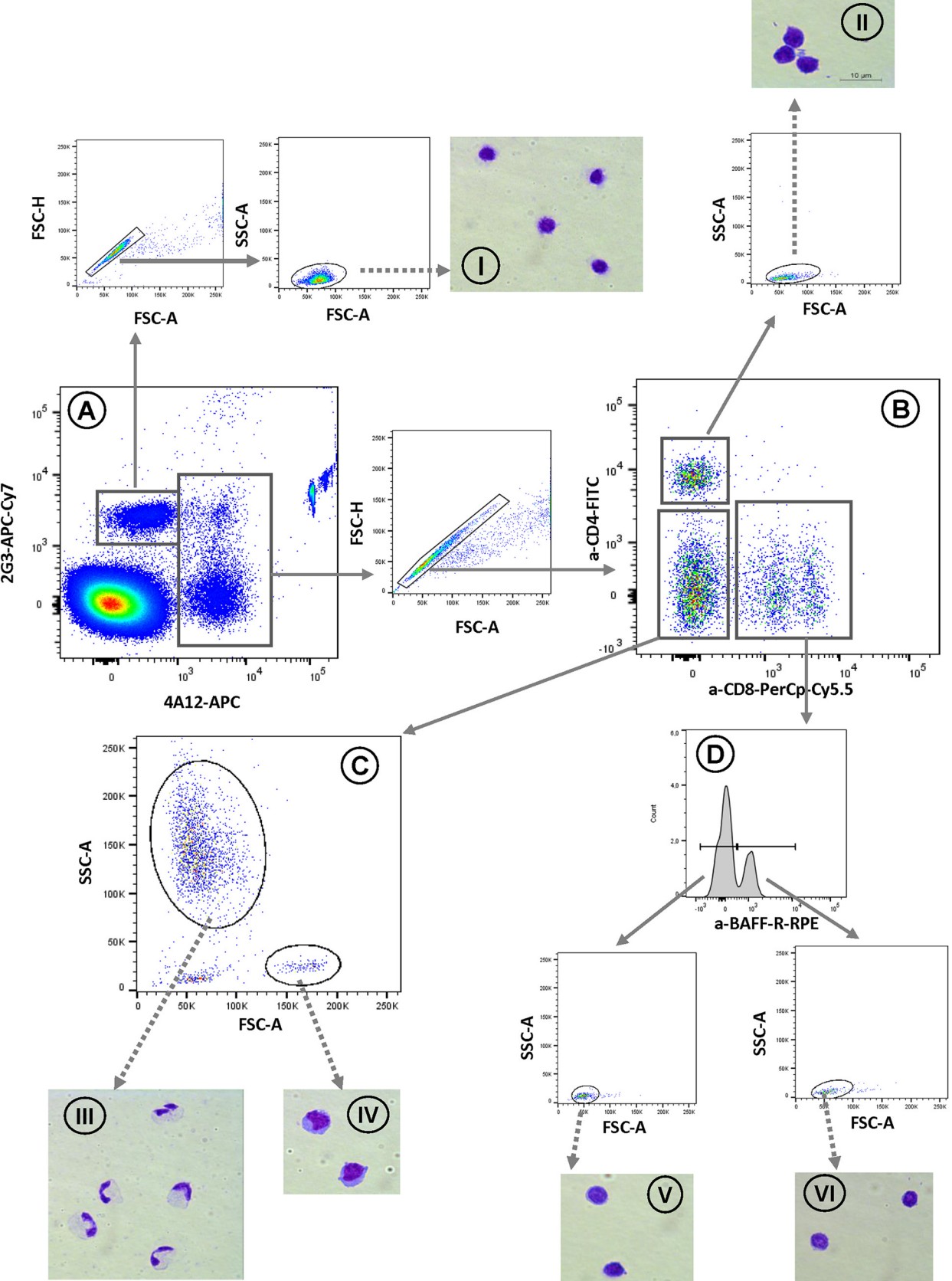

**FIG 1** Gating strategy. EDTA-blood was stained with 4A12-APC, anti-chBAFF-R (2C4)-PE, anti-duCD4-FITC, anti-duCD8-PerCp, and anti-2G3-APC-Cy7 in a no-lyse no-wash single-step one-tube procedure and analyzed by flow cytometry. Shown is the gating strategy for the separation of

The CD4/CD8 double negative population was further analyzed according to its scatter properties, which revealed two distinct cell populations as follows: a minor population with large cells (high FSC) and low granularity (low SSC) resembling monocytes and a major population with much smaller cells of high granularity resembling granulocytes.

To obtain absolute cell numbers, a bead-based method was applied and samples were directly stained in TruCount tubes. The highly fluorescent TruCount beads can be distinguished in every fluorescence channel, and we addressed them in an Fluorescein-Isothiocyanate/Peridinin-Chlorophyll-Protein Complex (FITC/PerCp) plot followed by a subsequent FSC/SSC plot to eliminate contaminating erythrocytes and evaluate the percentage of singlet and double sized doublet beads as described in detail in Seliger et al. (14).

To verify the specificity of the selected staining and gating strategy, we used the selected antibody combination to sort purify the respective subpopulations and perform a morphological analysis by microscopy of DiffQuick-stained cytospin preparations (Fig. 1). This revealed that cells, which our gating strategy addresses as CD4$^+$ T cells, CD8$^+$ T cells, and B cells were all of lymphocyte morphology. 2G3-stained cells showed a clear thrombocyte phenotype, and purification of 4A12-positive large cells with low granularity resulted in a highly pure monocyte preparation. Interestingly, cells purified from the granulocyte gate contained a majority of heterophils but also a distinct number of eosinophils. This can be attributed to the fact that granules from mallard eosinophils and heterophils both have rod-like granules and hence identical refraction properties, which impedes scatter-based discrimination. This is in contrast to chicken leukocytes, where granules of heterophils and eosinophils differ in density and shape and thus lead to differing scatter properties.

**(iii) Validation of the flow cytometry protocol by microscope-based analysis.** To validate the new automated differential blood count for the mallard, a subset of blood samples was analyzed in parallel by flow cytometry and microscope-based analyses. As different subsets of WBCs can be distinguished by each technique, the flow cytometry counts for granulocytes were compared to the combined total number of eosinophils and heterophils from the microscope-based analysis. Further, microscopy counts for lymphocytes were compared to the total number of B cells, CD4 T cells, and CD8 T cells from the flow cytometry-based WBC differential since B and T cells cannot be distinguished by light microscopy. The comparison of the results from the two techniques showed that the cell numbers measured with each method were within the same range (Fig. 2A). While the coefficient of variation (COV) was lower than 10% for all cell types when measured with flow cytometry (2.3% to 9.3%), the COV from the microscopy measurement ranged from 11.2% to 51.3% (Fig. 2B; see also Fig. S4 in the supplemental material).

**(iv) Evaluation of the effect of fixation on blood samples.** As staining and analysis of samples within hours upon blood withdrawal may not be possible under field conditions, we estimated the effect of fixation and storage on obtained cell numbers in mallards. The comparison of fresh and fixed samples showed that the COV for all technical triplicates, beside a single granulocyte count, was below 20%, with the vast majority of samples having a COV below 10%. When calculating the accuracy of the measurements after 2 and 7 days compared to the counts obtained at the day of sampling, all samples beside one B cell count, showed the expected accuracy between 70 and 130% (see Fig. S5 in the supplemental material). Hence, stabilization of duck blood samples for up to 7 days is possible.

**Field application of the differential blood count.** To obtain baseline blood cell count values and to investigate possible effects of age, sex, and avian influenza virus

**FIG 1** Legend (Continued)

thrombocytes and leukocytes (A), followed by doublet exclusion. (B) Leukocytes were classified as CD4 or CD8 single positive or CD4/CD8 double negative cells. (C) By means of their scatter properties, CD4/CD8 double negative cells were subdivided into granulocytes and monocytes. (D) CD8$^+$ leukocytes were classified into 2C4-negative T cells and 2C4-positive B cells. Initial thrombocyte and leukocyte discrimination (A) was followed by doublet exclusion. Fluorescence-based gates were always followed by FSC/SSC gating. FSC/SSC gates were used to sort purify the respective cell populations (purity, >95%) and prepare cytospin slides followed by DiffQuick staining for thrombocytes (I), CD4$^+$ T cells (II), granulocytes (III), monocytes (IV), CD8$^+$ T cells (V), and B cells (VI).

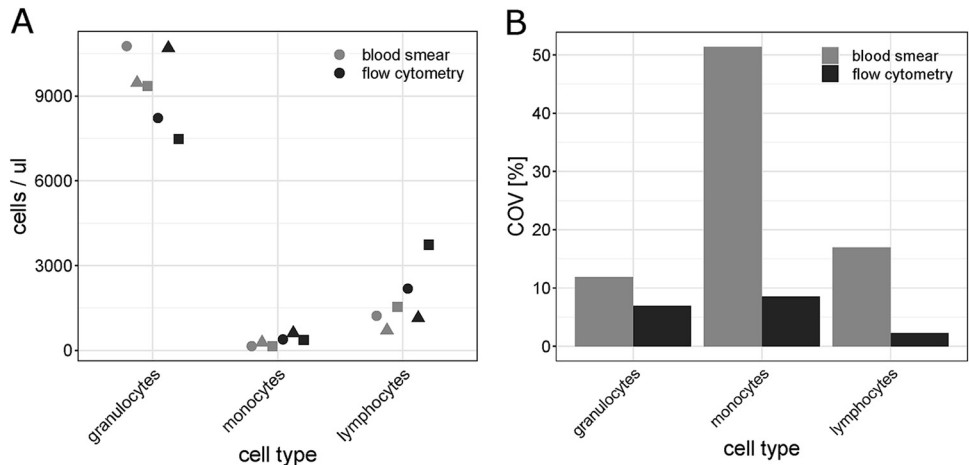

**FIG 2** Validation of the flow cytometry protocol. To validate the flow cytometry protocol, some samples were analyzed by flow cytometry as well as the microscopy-based technique. For the latter, the blood smears were subjected to a Wright-Giemsa staining and analyzed by microscope-based cell counting according to the modified Campbell method. The figure shows the absolute quantification based on a single sample from three individual ducks (A) and coefficient of variation (COV) in percentage estimated from five aliquots from a single EDTA-blood sample (B), for the WBC types that can be quantified using both methods. Flow cytometry counts for granulocytes are compared to the total number of eosinophils and heterophils in conjunction from the microscope-based analysis. Microscopy counts for lymphocytes are compared to the total number of B cells, CD4 T cells, and CD8 T cells from the flow cytometry-based white blood cell differential.

(AIV) infection on WBC counts in mallards, we applied our protocol in a population of wild mallards in southern Sweden.

**(i) Baseline information on WBC counts in mallards.** To determine baseline blood cell count values for mallards, we collected blood and fecal samples during autumn migration at Ottenby Bird Observatory.

We first determined the baseline WBC counts for mallards using the results from all individuals that tested negative for AIV ($n = 116$) (Table 2). The estimated mean and 95% credible interval (CrI) for all analyzed cell populations, as well as the observed mean and standard deviation (SD) for mallards grouped according to age and sex are visualized in Fig. 3 and Tables S1 and S2 in the supplemental material.

Interestingly, our analysis showed that age has a strong effect on WBC numbers in mallards. Based on the applied linear model, we found higher cell numbers of all analyzed leukocyte populations in male juvenile birds than in adult males (>95% certainty for monocytes and all lymphocytes and >75% certainty for granulocytes). Though less pronounced and restricted to lymphocytes, the effect was also observed in female birds (>95% certainty for CD4 T cells and total lymphocytes and >75% certainty for B cells and CD8 T cells, respectively). No age difference was observed for thrombocyte numbers (Fig. 3).

In addition, we identified an effect of sex on WBC counts in mallards. For thrombocytes and all lymphocyte populations, the applied linear model showed that juvenile male birds have higher cell numbers than juvenile female birds (>95% certainty for B cells, CD4 T cells, and total lymphocytes and >75% certainty for CD8$^+$ T cells and thrombocytes, respectively). The only effect in adult birds was seen for monocytes; here as an exception, the number in adult females was higher than in adult males (>75% certainty) (Fig. 3).

**TABLE 2** Number of mallards included in the study

| | AIV infection status[a] | |
|---|---|---|
| **Group** | **AIV−** | **AIV+** |
| Juvenile females | 44 | 23 |
| Juvenile males | 41 | 23 |
| Adult females | 12 | 4 |
| Adult males | 19 | 1 |

[a]AIV−, negatively tested for AIV infection by PCR, AIV+, positively tested for AIV infection by PCR.

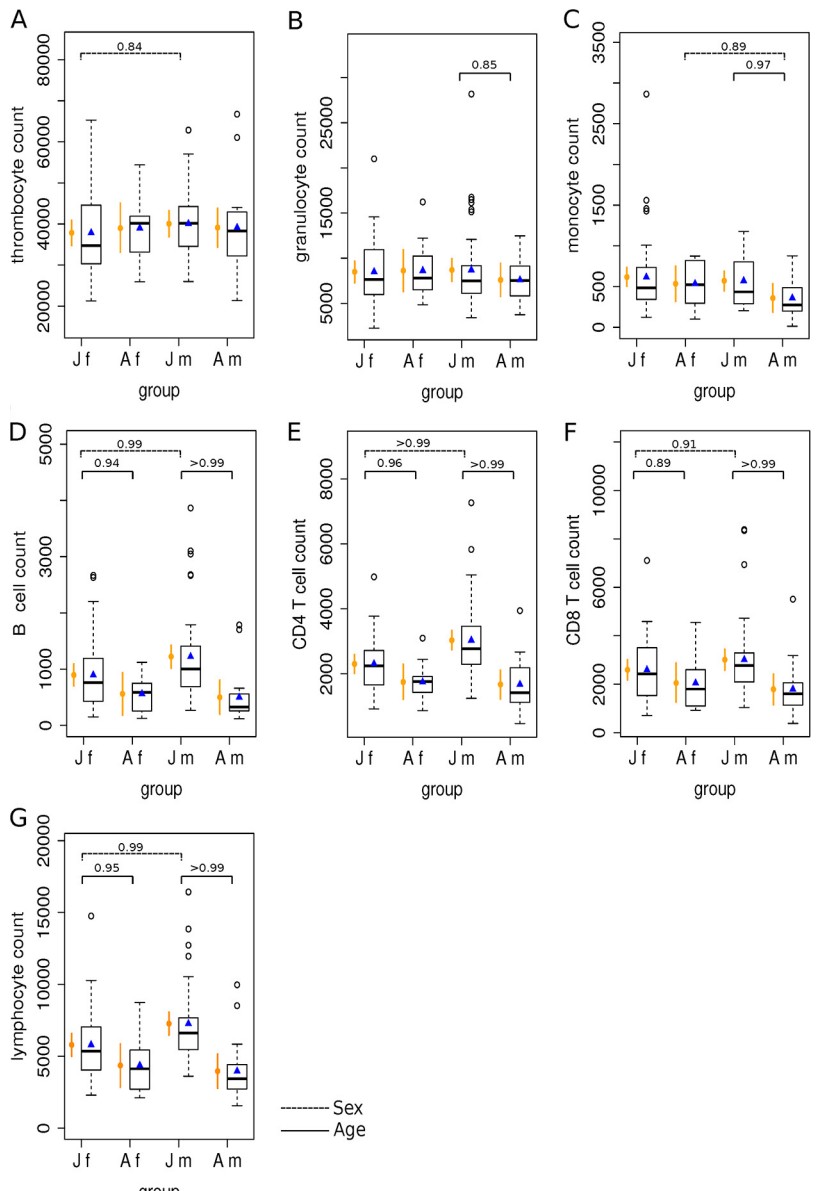

**FIG 3** Observed and estimated blood cell counts (cells/μL) for wild mallards grouped according to age and sex for thrombocytes (A), granulocytes (B), monocytes (C), B cells (D), CD4 T cells (E), CD8 T cells (F), and lymphocytes (G). Observed counts are presented as box plots. The horizontal line displays the median, the box includes the second and the third quantile, the whiskers include all values within the 1.5 interquartile range, dots represent outliers, and the mean is presented as a blue triangle. Estimated count values using a linear model are presented in orange, with dots displaying the estimated mean and vertical lines the respective 95% credible interval (CrI) for each group on the *x* axis. Values above brackets show the certainty that the mean in one group is larger than the mean of the other group. Certainty values of >0.8 are reported in the figure. The analysis and the simulation are based on *n* = 116 AIV-negative mallards, including *n* = 41 juvenile males (Jm), *n* = 44 juvenile females (Jf), *n* = 12 adult females (Af), and *n* = 19 adult males (Am).

Hence, generally, for most WBC populations, juvenile mallards have higher numbers than adults, and males have higher numbers than females.

**(ii) The effect of AIV infection on WBC counts in mallards.** To determine if AIV infection has an influence on WBC counts in mallards, we screened fecal samples from all wild individuals for AIV and compared the WBC counts in AIV-negative and AIV-positive birds. A total of 51 out of the 167 obtained mallard samples (30.5%) were tested positive for AIV. Among those were one adult male, four adult females, 23 juvenile males, and 23 juvenile females. As the number of AIV-positive samples among adult mallards was very

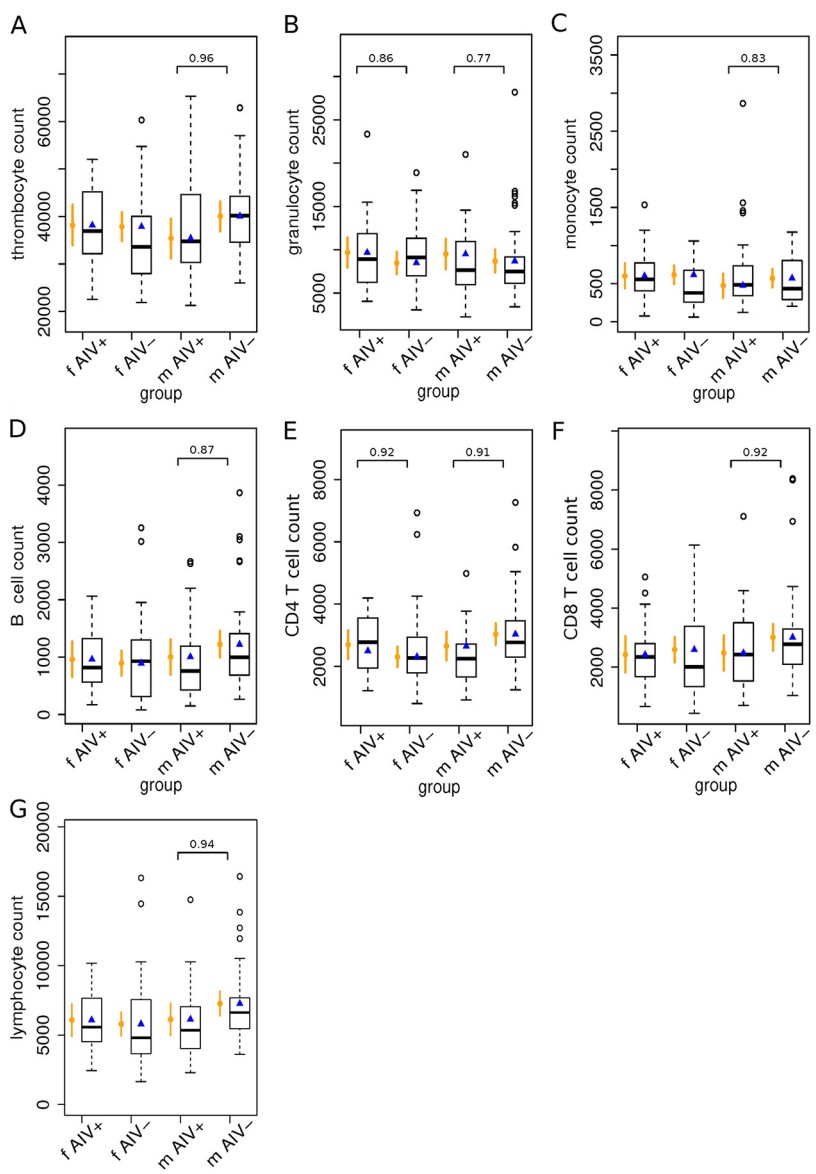

**FIG 4** Observed and estimated blood cell counts (cells/$\mu$L) for juvenile mallards grouped according to AIV infection state and sex for thrombocytes (A), granulocytes (B), monocytes (C), B cells (D), CD4 T cells (E), CD8 T cells (F), and lymphocytes (G). Observed counts are presented as box plots. The horizontal line displays the median, the box includes the second and the third quantile, the whiskers include all values within the 1.5 interquartile range, dots represent outliers, additionally the mean is presented as a blue triangle. Count values were estimated using a linear model and are presented in orange. Dots display the estimated mean, and vertical lines display the respective 95% credible interval (CrI) for each group on the *x* axis. The estimated count values were extracted from the simulated values from the posterior distributions of the group means of the linear model. Values above brackets show the certainty that the mean in one group is larger than the mean of the other. Certainty values of >0.8 are reported in the figure. The analysis is based on the observed blood cell counts of *n* = 131 juvenile mallards, including *n* = 23 AIV-positive males (mAIV+), *n* = 23 AIV-positive females (fAIV+), *n* = 44 AIV-negative males (mAIV−), and *n* = 41 AIV-negative females (fAIV−).

low, only blood cell counts from juvenile birds were used to determine whether AIV infection has an effect on WBC counts in mallards. Due to the shown sex effect on physiological WBC counts, we performed a separate analysis for females and males.

Strikingly, almost all WBC populations in males were lower in AIV-positive birds, while females showed no AIV-mediated decrease of WBC counts at all (Fig. 4; see also Table S3 in the supplemental material). The strongest reduction was found for thrombocytes of male birds with an estimated mean of 35,407 (95% CrI, 31,253 to 39,543) in

AIV-positive compared to 40,066 (95% CrI, 36,938 to 43,253) in AIV-negative birds (certainty, >95%) (see Table S4 in the supplemental material). In contrast to the reduced number of CD4 T cells in AIV-positive males, these cells were slightly increased in AIV-positive females.

In contrast, granulocyte counts were increased in AIV-positive males and females.

We further investigated if there was an association between the quantitative reverse transcriptase PCR (RT-qPCR) cycle threshold ($C_T$) value (a proxy for viral titer) and the WBC counts. None of the cell counts significantly correlated with the $C_T$ value ($N = 46$, $P > 0.05$, Spearman's rho = $-0.22$ to $0.11$) (see Fig. S6 in the supplemental material).

**(iii) Association of body condition with age, sex, and infection status.** We investigated if body condition differed between different groups of mallards in our study. Juvenile females and males differed in their body condition, with females having lower body condition than males (analysis of variance [ANOVA] [$n = 116$], F[3, 88] = 7.99, $P = <$ 0.001, and Tukey's honestly significant difference [HSD] *post hoc* test $t_{Jf-Jm}$ = 4.667, $p_{Jf-Jm} <$ 0.001) (see Fig. S7A in the supplemental material). In contrast, there was no difference between infected and noninfected birds (two-sided *t* test; females [$n = 64$], $t[50.51]$ = 1.01, $P = 0.319$, and males [$n = 67$], $t[46.38]$ = 0.63, $P = 0.532$) (Fig. S7B). Likewise, no significant correlation was detected between the $C_T$ value and body condition of the birds ($N = 46$, $P > 0.05$, Spearman's rho = $0.05$ to $0.33$) (Fig. S7). As no clear pattern was observed in body condition between infected and noninfected birds, this factor was not included in the main analysis.

## DISCUSSION

Automated techniques developed for the determination of WBC counts, and differentials are routinely used in human and veterinary medicine. Their application in avian medicine has, however, been hindered by unique properties of avian blood cells (20). In contrast to mammalian erythrocytes, avian erythrocytes are nucleated and resistant to common lysis procedures involved in WBC differentials (21). In addition, while mammalian platelets can be easily distinguished from the WBCs by their characteristic structure, this is not the case for avian thrombocytes (the equivalent to mammalian platelets) (20). Therefore, labor-intensive techniques, such as microscopy and manual counting are still routinely used in avian medicine and research (22, 23). In this study, we established an automated differential blood count for mallards using a combination of cell-specific monoclonal antibodies and flow cytometry, as we did previously in chicken (14). The simple (no-lyse no-wash single-step one-tube) flow cytometry-based technique allows researchers and clinicians a much faster analysis of a large number of blood samples within hours. By evaluating the same samples using microscopy and manual counting alongside the automated differential blood count for mallards, we demonstrate that the cell numbers measured with each method are within the same range and that flow cytometry is the most precise technology so far for evaluating mallard blood samples.

Currently, our protocol allows for the quantification of mallard thrombocytes, granulocytes, monocytes, B cells, CD4 T cells, and CD8 T cells. We are aware of the fact that for an unequivocal identification of cytotoxic and helper T cells, staining for CD4 and CD8 should be combined with T cell receptor (TCR) or CD3 staining, as, e.g., in chicken, some NK cells also express CD4 and CD8 (24, 25). But unfortunately, to date, we lack antibodies against mallard TCR or CD3. However, in mammals and chicken, NK cells are larger than nonactivated T cells and, hence, show slightly different FSC properties (14, 26). As our protocol includes additional FSC/SSC gating for small lymphocytes, we consider it valid to address the identified CD4+ and CD8+/BAFF-R cells as CD4 T cells and CD8 T cells. If more cell-specific MAbs become available, the protocol can easily be adjusted and could then, e.g., also include the quantification of natural killer cells or allow for the discrimination between heterophils and eosinophils, which in contrast to chickens, cannot be discriminated according to their scatter properties.

As we expect that one of the main applications of our protocol will be immune status monitoring of wild mallards under field conditions, we evaluated the effect of stabilization

of whole-blood samples using a fixative reagent. Our results show that stabilized samples can be stored up to a week without any major changes occurring in the WBC composition. This means that samples can be collected in the field, stabilized, and shipped to a reference laboratory for further analysis. Throughout our project, we did, however, notice that duck B cells are very fragile. They often did not survive fixation and shipment, lost typical scatter properties, and were found among the population of small cells, which in unfixed samples is partially composed of dead cells. However, as long as only debris and no intact cell is excluded from analysis, getting a differential count including B cells from fixated samples is still possible.

To monitor the immune status of mallards and to detect deviations from a "normal" state, baseline information for WBC counts from healthy mallards is required. So far, only a limited number of studies reporting WBC counts from mallards are available, and they often report results from a small number of individuals due to the quantity limitations of the labor-intensive microscopy technique (12, 27–29). We therefore determined the baseline information for WBC counts from wild AIV-negative mallards using the newly developed flow cytometry protocol.

We compared the WBC counts from AIV-negative mallards with counts from other bird species where flow cytometric analysis methods have been established. While the mean count for most mallard WBC populations was within the reported range for turkeys (*Meleagris gallopavo*) (30) and similar to the mean counts in chicken (14), we observed a wider range of granulocyte counts and a lower number of monocytes in the mallards. This observation may be explained by higher individual variation in wild populations compared to the domesticated turkeys or an ongoing infectious disease (other than AIV) or a small injury (which was screened for but may have remained undetected during catching) in the mallards.

The high individual variation observed in the wild mallards was expected, as factors that may impact WBC counts (i.e., life history and migratory behavior) could not be determined for each bird. In fact, studies of wild animals are complex due to the large number of intrinsic and extrinsic factors that cannot be accounted for, unlike experimental studies. As sex and age affect the functions of the immune system (reviewed in references 17 and 18), we accounted for these factors in our study. To further account for individual variation in host physiology, we calculated the body condition of each individual and compared it between mallards of different age, sex, and infection status. Body condition differed between juvenile males and females but not between AIV-positive and -negative birds. As body condition mainly correlated with sex, we did not include this factor in our analysis. Further studies are required to investigate the effect of intrinsic and extrinsic factors on WBC counts in mallards.

The immune system is known to experience profound changes with aging (reviewed in reference 17), and indeed, we found higher numbers of all types of lymphocytes in juvenile birds compared to adult mallards. In males, we further observed a higher number of monocytes in juveniles than in adults. The observed decrease of B cells and T cells with age in mallards is reflective to age-related atrophies of the avian primary lymphoid organs, the bursa of Fabricius and the thymus (31, 32).

As a growing body of literature shows that females and males differ in their immunity (reviewed in reference 18), we tested for a potential sex effect on WBC counts. Interestingly, we observed a strong effect of sex in the baseline blood cell count in juvenile mallards with males having higher lymphocyte counts than females. In adult mallards, the only observed difference was for monocytes, where females had a slightly higher number of cells than males. Our results for adult mallards are in line with a previous study that did not find major sex differences between the ratio of WBCs in adult mallards (12). It is important to note, however, that the blood samples used in our study were collected outside the breeding season. As sex hormones play an important role in the transcriptional regulation of the immune system (33), sex differences are likely to be most pronounced during the breeding season (34). While a detailed investigation of seasonal differences of WBC counts in mallards was outside the scope of this

study, our automated differential blood count for ducks facilitates such studies in the future.

Finally, we tested whether LPAIV infection had an effect on the WBC counts in wild mallards. Thirty-five percent of juvenile mallards in this study tested positive for AIV infection, including 35% of the male juveniles and 36% of the female juveniles, thus giving no indications that juvenile females and males differ in their exposure to AIV. While determining the subtype of the AIV-positive samples was outside the scope of this study, a long-term study monitoring AIV at the same location detected a wide range of AIV subtypes in mallards, of which all were low pathogenic (35). It is, therefore, likely that the AIV-positive samples detected in this study were of low pathogenicity.

Notably, we observed lower numbers of all types of lymphocytes in AIV-positive males. A reduction of lymphocytes in the blood (lymphocytopenia) is common during subclinical infections such as the common cold or the flu and has been found to be an early and reliable laboratory finding of adult influenza A in humans (36, 37). Likewise, evidence of lymphocytopenia has been observed during asymptomatic infection in the Marburg virus reservoir, the Egyptian rousette bat (38). At least in male mallards, lymphocytopenia may thus be an indicator of subclinical AIV infection. Interestingly, no signs of lymphocytopenia were observed in infected female mallards. Instead, the number of CD4 T cells was higher in AIV-positive females than in AIV-negative females. This observation may be a result of the effect of sex hormones on the activity and distribution of CD4 T cell subsets in females and males (18). To further explore this possibility, we propose to quantify sex hormones and to characterize the T helper 1/T helper 2 cytokine response in infected mallards in combination with our protocol in future studies. Further, it would be helpful to screen the individuals for more than one pathogen in follow-up studies.

The only cell population that had a higher abundance both in AIV-positive females and males was granulocytes. Avian granulocytes include heterophils, a counterpart to the mammalian neutrophil, as well as basophils and eosinophils. In birds, heterophils are the most numerous granulocytes in circulating blood, and they play a critical role in the initial replication and dissemination of highly pathogenic AIV (39). To definitely determine if the observed increase in granulocyte counts in mallards is due to an increase in heterophils, further development of the technique to differentiate between the different types of granulocytes is required, but due to the reported low number of eosinophils and basophils, it is very likely. Still, our study indicates that elevation of blood circulating granulocytes could be a sign of AIV infection in, at least, juvenile mallards.

Finally, a decrease of thrombocytes was observed in AIV-infected males. While thrombocytes have long been known to be involved in hemostasis and wound repair, an increasing body of literature now shows that they have roles in inflammation and immunity (40). In humans and chicken, severe illness with high pathogenic AIV have been associated with thrombocytopenia (decrease in thrombocytes) (41, 42). Our results and protocol may help to improve our understanding of the importance of thrombocytes in AIV infection in mallards.

Cycle threshold ($C_T$) values obtained from RT-qPCR analyses indicate how much viral genetic material is in a sample and can thus be used as a proxy for viral titer. Using the $C_T$ value of all AIV-positive juveniles, we investigated whether number of WBCs correlate with viral titer in mallards. Overall, none of the cell counts strongly correlated with the $C_T$ value. In this study, $C_T$ value was thus not a good indicator for the strength of immune response in individual birds. When looking at individual birds, the five birds with the lowest $C_T$ value (indicating higher viral quantity) had relatively low CD8 T cell, B cell, and lymphocyte counts. This suggests that individuals with a particularly high viral titer may show stronger signs of lymphocytopenia. Additional studies are required to better understand the effect of viral titer on WBC counts in mallards.

To our knowledge, this is the first high-throughput technology for estimating WBC counts in mallards. Our study provides the first reference values for absolute cell numbers for a wide range of WBCs in wild mallards of different age, sex, and LPAIV infection status. As blood samples can easily be retrieved from wild as well as domesticated

mallards, our protocol can be used for a wide range of applications, including monitoring immune status and diseases. As stress has been shown to alter the composition of WBCs in birds (43), the protocol can further be used to improve our understanding of the role of stress in wild populations, for example in relation to changing environments or breeding seasons. Due to the demonstrated cross-reactivity of some components between Galliformes and Anseriformes, it is likely that our protocol can also be applied to closely related species, and as the described method to generate a species-specific pan-leukocyte marker is applicable for every bird species, it can be modified for a wide range of species.

In conclusion, we present a technique that enables simple, rapid, and accurate high-throughput immune status monitoring in an important host of avian influenza virus (AIV), the mallard. By applying our protocol in wild individuals from a well-studied population in southern Sweden, we present the first absolute count baseline information for a number of WBC populations in this species. We show that age has an effect on the baseline blood cell count in mallards, as does sex in juvenile mallards. Interestingly, our results show that naturally infected mallards do not exhibit a strong immune response to LPAIV infection. This result is in agreement with those from previous studies in mallards and can likely be explained by the long-term coevolution between mallards and LPAIV (6, 22). Nevertheless, we identified a phenotype that may be used to identify AIV-infected mallards, namely, an increased number of granulocytes in females and males as well as a decrease in lymphocytes and thrombocytes and in males. This could be tested experimentally with a single pathogen. Our results give important insight to the interplay of LPAIV with the immune system of mallards and provide a tool to further investigate the immune response in an important reservoir host of zoonotic viruses.

## MATERIALS AND METHODS

**Establishment of the mallard differential blood count protocol. (i) Collection and processing of blood samples.** We used surpluses from diagnostic blood samples from captive male and female mallards (aged between 6 weeks and 2 years) housed in outdoor aviaries at the Max Planck Institute for Ornithology in Radolfzell, Germany. The individuals were born in captivity but were of the third or fourth generation of wild mallards. Blood samples were collected from the wing vein (*V. cutanea ulnaris*) and immediately placed in K3-EDTA-coated tubes (Sarstedt, Nümbrecht, Germany). When indicated, duck peripheral blood mononuclear cells (PBMCs) were separated by density gradient centrifugation on Biocoll solution (Merck, Darmstadt, Germany). Fixation of blood samples was obtained by the addition of TransFix reagent (Cytomark, Buckingham, UK) according to the manufacturer's instruction. Unfixed blood samples were kept at room temperature (RT) and processed within 4 h after blood collection. To prepare blood smears, we placed 5 $\mu$L of whole blood in EDTA on a microscope slide and used the wedge smear technique as described in Seliger et al. (14).

**(ii) Antibodies.** Our method is based on a combination of available duck-specific MAbs (anti-duCD8-1 and anti-duCD4-1), a new cross-reacting chicken MAb (anti-chBAFF-R, 2C4), and two newly generated duck-specific MAbs (2G3 and 4A12).

Mouse anti-duck CD4 (clone duCD4-1; IgG1) and mouse anti-duck-CD8$\alpha$ (duCD8-1; IgG2b) were generated as described previously (15) and are commercially available from Bio-Rad GmbH (Puchheim, Germany).

For the generation of an anti-chBAFF (B cell activating factor of the tumor necrosis factor [TNF] family) receptor antibody, we prepared a chBAFF-R-GFP-expressing plasmid by amplifying the chBAFF-R (TNFRSF 13C) sequence (GenBank accession number NM_001037828; GeneID 417983) from bursa cDNA using forward primer 5'ATGCAGGAGCGCTCGGCCATGGC-3' and reverse primer 5'GCAGTCTCTCCTCACCCATA CACTC-3'. The PCR product was ligated into pcDNA 3.1/CT-GFP-TOPO (Thermo Fisher Scientific, Waltham, USA), and HEK293 cells were stably transfected and selected for green fluorescent protein (GFP) fluorescence. A BALB/c mouse was immunized three times with HEK293_BAFF-R-GFP. Murine spleen cells were fused to SP2/0-Ag14 cells, and supernatants of resulting hybridomas were tested by flow cytometry on transfected and untransfected HEK293 cells and primary chicken cells. The selected hybridoma 2C4 is an IgG1 antibody. Figure S3 in the supplemental material demonstrates specific binding of 2C4 to mallard B cells.

To generate additional monoclonal antibodies against mallard WBCs, a BALB/c mouse was immunized three times intraperitoneally with $1 \times 10^8$ mallard PBMCs in phosphate-buffered saline (PBS) (authorized by the Regierung von Oberbayern; experimental-number 55.2-1-54-2532.0-60-2015). Three days after the second boost, murine spleen cells were fused to SP2/0-Ag14 hybridoma cells. Specificity of resulting hybridomas was examined by flow cytometry using undiluted hybridoma supernatants and goat anti-mouse-IgG-FITC (Sigma-Aldrich, USA; 1:200) on mallard PBMCs.

We identified MAb 4A12 (IgG1), which recognizes mallard lymphocytes (T cells and B cells), granulocytes, and monocytes and thus can be used as a duck pan-leukocyte marker (see Fig. S1 in the supplemental material). We further identified MAb 2G3, (IgG1) which strongly binds to duck thrombocytes (see Fig. S2A in the supplemental material).

Selected MAbs were purified by affinity chromatography using Protein G Sepharose, Fast flow (Merck, Darmstadt, Germany), and conjugated to fluorescein isothiocyanate (FITC), R-PE, PerCP-Cy5.5, APC, and APC-

Cy7 using the LYNX rapid antibody conjugation kit (Bio-Rad AbD Serotec GmbH, Puchheim, Germany) following the manufacturer's instructions (Table 1).

**(iii) Flow cytometry and cell sorting.** All flow cytometric measurements were performed within 90 min post staining using a FACSCanto II (Becton, Dickinson, Heidelberg, Germany), and data were analyzed using FACSDiva (Becton, Dickinson, Heidelberg, Germany) and FlowJo (FlowJo LLC, OR, USA) software. For absolute quantification, at least 10,000 Trucount beads were detected in each sample. The absolute numbers of individual cell populations were calculated using the following equation:

$$\frac{\text{cells counted}}{\text{beads counted}} \times \frac{\text{total number of beads per tube}}{\text{blood volume per tube } (\mu\text{L})} = \frac{\text{absolute cell count}}{\mu\text{L blood}}$$

Purification of cell populations from stained full blood samples was performed using a FACSArialIIu with FACSDiva software.

**(iv) Cytospins.** To obtain cytospin specimens from sorted cell samples, a 200-$\mu$L cell suspension was centrifuged on a glass slide, air dried, fixed with methanol, and subjected to Diff-Quik staining (Medion Diagnostics, Düdingen, Switzerland).

**(v) Flow cytometry-based WBC quantification protocol for mallard blood samples.** Staining of full blood samples for flow cytometric analysis was essentially performed according to the no-lyse no-wash single-step one-tube described for chicken blood by Seliger et al. (14). Briefly, directly conjugated antibodies were all diluted together in staining buffer (PBS, pH 7.2, 1% bovine serum albumin [BSA], 0.1% NaN$_3$). Twenty microliters of the EDTA-blood was diluted with 980 $\mu$L staining buffer, and 50 $\mu$L of the diluted blood sample were then mixed with 20 $\mu$L of the antibody mixture in a BD Trucount tube (BD, Heidelberg, Germany). After 45 min of incubation in the dark at RT, 300 $\mu$L staining buffer was added, and samples were kept on ice until measurement. In order to reduce measurement errors and to ensure precise pipetting, reverse pipetting was used for all steps.

**(vi) Microscope-based WBC quantification for mallard blood samples.** To validate our flow cytometry protocol, blood samples from three individuals were analyzed in parallel by flow cytometry and microscope-based analyses, and the obtained absolute numbers of WBC subpopulations were compared. Staining of nonfixed, airdried blood smears and analyses by oil immersion light microscopy were performed by PendlLab, Switzerland, a commercial laboratory specialized in hematology, cytology, and histology in birds and reptiles. Absolute leukocyte and thrombocyte numbers were generated by the modified (44) estimation method (45) described in reference 14.

While the flow cytometry protocol can distinguish thrombocytes, granulocytes, monocytes, B cells, CD4$^+$ T cells, CD8$^+$ T cells, and lymphocytes (sum of B cells and CD4$^+$ and CD8$^+$ T cells), the microscopical analyses discriminates heterophils, eosinophils, basophils, monocytes, and lymphocytes. To allow for comparison of both techniques, we added up the number of eosinophils and heterophils from the microscope-based analysis to compare granulocyte numbers, and the number of B cells, CD4$^+$ T cells, and CD8$^+$ T cells from the flow cytometry-based WBC differential to get comparable measures for lymphocytes.

To compare the specificity of our flow cytometry protocol with the microscopy technique, a fresh EDTA-blood sample was analyzed in five replicates using both methods, and mean, standard deviation, and coefficient of variation (COV) for granulocytes, lymphocytes, and monocytes were determined for each method separately.

**(vii) Blood sample stabilization.** To establish the staining protocol, all EDTA-blood samples were stained and analyzed within hours upon blood withdrawal. As this may not be possible under field conditions, we estimated the effect of fixation and storage on obtained cell numbers in mallards. Blood samples from four birds were therefore split into three aliquots and either analyzed immediately after blood withdrawal or stabilized and analyzed after 2 and 7 days of storage. For stabilization, EDTA-blood samples were mixed with TransFix reagent (Fisher Scientific, Schwerte, Germany) according to the manufacturer's instructions and stored at 4 to 8°C. Until fixation, samples were stored on a roll mixer at RT.

**Field application of the differential blood count. (i) Sample collection in a wild mallard population.** To test the WBC count applicability under field conditions, we collected blood samples during autumn migration at Ottenby Bird Observatory (56°13′N, 16°27′E), an important stopover site for migrating birds in Northern Europe. At the study site, wild mallards are captured daily from April to December in a duck trap, which houses a few sentinel mallards (for details, see 46). Out of the 176 collected blood samples, six samples were excluded due to failed anti-CD8 staining (likely caused by CD8 polymorphisms), and samples from three mallards were excluded due to low quality (thrombocyte cell counts of <20.000, which suggests blood clotting).

From all birds, age, sex and body condition were determined, and AIV infection status was analyzed in cloacal swabs taken at the time of blood sampling. The sex and age of all mallards included in this study were determined based on the guide from Andersson et al. (47). After age determination, ringing, sexing and sampling, all ducks were released in the trap's vicinity. For this study, we analyzed blood and fecal/cloacal samples from 176 mallards captured between 26 October and 20 November 2019.

Blood and fecal sampling was approved by Linköpings djurförsöksetiska nämnd (2017-1068).

**(ii) Avian influenza virus screening of fecal samples.** To collect fecal or cloacal swabs for virus detection, the ducks were placed in a single-use cardboard box for one to 3 h. Fecal material in the boxes was collected using a sterile, cotton tipped applicator. If no feces were obtained, cloacal swabs were taken. Material was immediately placed in virus transport medium (Hanks' balanced salt solution containing 0.5% lactalbumin, 10% glycerol, 200 U/mL penicillin, 200 $\mu$g/mL streptomycin, 100 U/mL polymyxin B sulfate, 250 $\mu$g/mL gentamicin, and 50 U/mL nystatin; Sigma) and stored at −80°C within 1 to 4 h after collection.

Detection of influenza A virus (IAV) in fecal samples was performed as described previously (48). Briefly,

viral RNA was isolated using the MagNA Pure 96 nucleic acid purification system (Roche, Mannheim, Germany) and MagNA Pure 96 DNA and viral nucleic acid large volume kit (Roche) following manufacturer's recommendations. Prior to extraction, the virus transport medium samples were diluted 1:4 with PBS. The samples were examined for viral RNA using a quantitative reverse transcriptase PCR (RT-qPCR) assay targeting the influenza matrix gene with the One-Step RT-PCR kit (Qiagen, Hilden, Germany), as described previously (49). Samples were considered to be IAV positive when viral RNA was detected within 40 amplification cycles.

**(iii) Statistics for the effect of age, sex, and AIV infection on WBC counts.** To determine baseline blood cell count values for mallards, and to investigate possible effects of age, sex, and AIV infection on WBC counts, we used linear models in a Bayesian framework. For these analyses, the mallards were divided into four groups (juvenile males [Jm], juvenile females [Jf], adult females [Af], and adult males [Am]).

To investigate a possible influence of age and sex on the numbers of WBC subpopulations, a linear model with the total cell counts as dependent variable and age, sex, and the two-way interaction of age and sex as fixed factors was used. All 116 AIV-negative individuals were used for this analysis (Table 2).

The influence of an AIV infection on WBC counts was analyzed with a linear model with the total cell count as dependent variable and sex, AIV infection, and the two-way interaction of sex and AIV infection as fixed factors. As the majority of the AIV-positive mallards were juveniles ($n$ Jm = 23, Jf = 23) (Table 2), we only included juveniles in this second model. Both models were run using the function "lm" of the package "stats" in R software 3.6.1 (50).

To obtain the posterior distribution, we simulated 10,000 values from the joint posterior distribution of the model parameters using the function "sim" of the package "arm" (51). The means of the simulated values from the joint posterior distribution of the model parameters were used as estimates, and the 2.5% and 97.5% quantiles as lower and upper limits of the 95% credible intervals (CrI).

For pairwise comparison between different groups, we calculated the posterior probability (certainty) of the hypothesis that the group mean of one group is larger than the group mean of the other group. We considered an effect to be significant if the certainty of being larger was >0.95 and an effect to be a trend if the posterior probability of being larger was >0.75.

We used a Spearman rank correlation analysis to investigate if the viral RT-qPCR $C_T$ value correlated with the WBC count of the individual birds.

**(iv) Association of body condition with age, sex, and infection status.** To investigate if body condition (body weight/wing length) differed between mallards of different age and sex, we performed an ANOVA and a Tukey *post hoc* test for pairwise means comparison. To test if body condition differed between infected and noninfected mallards, we used a two-sided $t$ test. To test if the viral RT-qPCR $C_T$ value correlated with the body condition of the individual birds, we further used a Spearman rank correlation analysis.

## SUPPLEMENTAL MATERIAL

Supplemental material is available online only.
**SUPPLEMENTAL FILE 1**, PDF file, 1 MB.

## ACKNOWLEDGMENTS

We are thankful to Daniel Zuñiga, Wolfgang Fiedler, and Erik Kleyheeg for their help collecting diagnostic samples and to Anna Moulin and Larissa Simulik for their help collecting samples from wild mallards. We are grateful for statistical support from Fränzi Korner-Nievergelt from Oikostat www.oikostat.ch/.

This study was generously supported by Martin Wikelski at the Max Planck Institute for Animal Behavior. This is contribution no. 330 from Ottenby Bird Observatory.

We have no competing interests.

E.J., E.W., I.M., J.O., J.W., R.K., and S.H. conceived the ideas and designed methodology; E.J., E.W., I.M., B.S., M.K., J.O., and S.H. collected the data; S.H. contributed unpublished, essential data or reagents; E.J., E.W., and S.H. analyzed the data and led the writing of the manuscript. All authors contributed critically to the drafts and gave final approval for publication.

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
