## [Reviewer comments · Microbiology Spectrum]

Microbiology Spectrum

Evaluating effects of AIV infection status on ducks using a flow cytometry based differential blood count

Elinor Jax, Elena Werner, Inge Müller, Beatrice Schaerer, Marina Kohn, Jenny Olofsson, Jonas Waldenström, Robert Kraus, and Sonja Härtle

Corresponding Author(s): Elinor Jax, Max-Planck-Institut für Verhaltensbiologie

Review Timeline:

Submission Date:	October 26, 2022
Editorial Decision:	February 10, 2023
Revision Received:	April 8, 2023
Accepted:	May 25, 2023

Editor: Rafael A. Medina

Reviewer(s): The reviewers have opted to remain anonymous.

Transaction Report:

DOI: <https://doi.org/10.1128/spectrum.04351-22>

February 10, 2023

Dr. Elinor Jax
Max Planck Institute of Animal Behaviour
Am Obstberg 1
Radolfzell
Germany

Re: Spectrum04351-22 (Evaluating effects of AIV infection status on ducks using a flow cytometry based differential blood count)

Dear Dr. Elinor Jax:

Thank you for submitting your manuscript to Microbiology Spectrum.

Please answer the reviewer's questions and add data about the health status monitoring if available.

Link Not Available

Sincerely,

Alison Sinclair

Senior Editor, Microbiology Spectrum

E-mail: spectrum@asmusa.org

Reviewer comments:

Reviewer #1 (Comments for the Author):

A method is described and validated to analyze WBC counts in field samples from wild mallards. The method is suitable to examine samples until up to 7 days, if samples had been stabilized immediately after sampling. Several strata were compared including gender, age and productive LPAIV infection and found to have an effect on WBC counts. LPAIV infection was associated with lymphocytopenia and thrombocytopenia in juvenile male mallards.

The study provides interesting and original data which closes gaps in our knowledge on LPAIV infections in wild waterbirds. The paper is well written and methodologically sound and transparent. Only the concept of "health monitoring" remains little specific and may even be misleading. The understanding after reading the paper is that the mallards sampled here, even if they turned out LPAIV positive, were clinically healthy. The definition of what is considered "healthy" is missing here. Therefore, expressions as in line 75 or 78 (health indicator) and elsewhere should be re-considered.

Just one further remark on possible improvements:

175 - When comparing WBC counts by FACS versus light microscopy the fact that no lymphocyte differentiation (B/T) is possible by microscopy is not mentioned.

Reviewer #2 (Comments for the Author):

This study by Jax et al. proposes a novel method for quantifying white blood cells in mallards using flow cytometry with the potential to implement this technique for samples collected in the field. The development of reagents, such as monoclonal antibodies that recognize leukocytes and thrombocytes originating from wild mallards represents a substantial amount of work and is a welcome addition to the understudied field of immunology of wild avian reservoirs. The gating strategy used to classify the cell populations using flow cytometry appears sound and reproducible and was evaluated relative to manual differential WBC counts using microscopy. The authors assess whether WBC numbers in mallards were effected by age, sex and infection with low pathogenic avian influenza. Age- and sex-specific differences in WBC numbers were detected which aligned with results of prior studies. Influenza infection did not correlate with any significant differences in WBC counts for which the authors suggest co-evolution between host and pathogen may be responsible. This is a timely paper, and the need for high throughput methods for quantifying blood cells that indicate health status of wild birds is compelling, especially against the backdrop of the current avian influenza virus outbreaks.

Below are two suggested revisions:

- The authors might consider framing some hypotheses regarding WBC and the effects of age, sex and infection with influenza virus. This would be a useful addition to the Introduction. For instance, how would co-evolution impact host and viral factors? And is this similar to relationships observed for immune cells of other wild reservoirs such as bats and rats. There's little doubt that this is first and foremost a method development paper. However, the field testing of the method on mallards from Ottenby is an important aspect of the study which therefore should be framed with some predictions about the relationship between blood cells, demographic and infection ecology.
- Would including Ct values from the influenza rRT-PCR provide useful context for understanding viral titer and perhaps where infected individuals are in the disease course? The lack of overall signal in WBC for infected juveniles might be confounded by the fact that early and late stages of infection are associated with different WBC profiles. Would also be useful to know if the IAV-positive mallards had sero-converted and therefore adaptive immunity had been mobilized. However, running serological assays to screen for antibodies is non-trivial and not vital to the outcome of the conclusions. I leave this at the discretion of the authors.
- Including any additional data on host physiology, such as body condition, would be helpful to interpret the age, sex and infection status results. Studies of wild animals are complex due to the huge number of intrinsic and extrinsic factors that cannot be accounted for, unlike experimental studies. For example, the life-history and migratory behavior of each mallard cannot be known, however, these are traits that may impact WBC as a measure of physiological health. Including this data or a statement on the ecological and physiological traits that were not measured in this study would be helpful for interpretation of the results.

Staff Comments:

Preparing Revision Guidelines

Please return the manuscript within 60 days; if you cannot complete the modification within this time period, please contact me. If you do not wish to modify the manuscript and prefer to submit it to another journal, please notify me of your decision immediately so that the manuscript may be formally withdrawn from consideration by Microbiology Spectrum.

Response to Reviewers (manuscript Spectrum04351-22)

Reviewer #1:

- 1. Reviewer's comment:** A method is described and validated to analyze WBC counts in field samples from wild mallards. The method is suitable to examine samples until up to 7 days, if samples had been stabilized immediately after sampling. Several strata were compared including gender, age and productive LPAIV infection and found to have an effect on WBC counts. LPAIV infection was associated with lymphocytopenia and thrombocytopenia in juvenile male mallards. The study provides interesting and original data which closes gaps in our knowledge on LPAIV infections in wild waterbirds. The paper is well written and methodologically sound and transparent.

Response: Thank you very much for your valuable feedback on our study.

- 2. Reviewer's comment:** Only the concept of "health monitoring" remains little specific and may even be misleading. The understanding after reading the paper is that the mallards sampled here, even if they turned out LPAIV positive, were clinically healthy. The definition of what is considered "healthy" is missing here. Therefore, expressions as in line 75 or 78 (health indicator) and elsewhere should be re-considered.

Response: Thank you for pointing this out. We agree that "health" is a complex concept, in particular in a study concerning a reservoir species that appear clinically healthy after an infection with a virus for which they share a co-evolutionary history. We have therefore removed the term "health" from most places in the manuscript, and replaced it with more appropriate terms such as cellular immune status.

- 3. Reviewer's comment:** 175 - When comparing WBC counts by FACS versus light microscopy the fact that no lymphocyte differentiation (B/T) is possible by microscopy is not mentioned.

Response: Thank you for spotting this. We have added a sentence to the result section explaining what subtypes can be detected by each method.

Changes implemented: MS lines 170-175: "As different subsets of WBCs can be distinguished by each technique, the flow cytometry counts for granulocytes were compared to the combined total number of eosinophils and heterophils from the microscope-based analysis. Further, microscopy counts for lymphocytes were compared to the total number of B cells, CD4 T cells and CD8 T cells from the flow cytometry based WBC differential since B and T-cells cannot be distinguished by light microscopy."

Reviewer #2:

- 1. Reviewer's comment:** This study by Jax et al. proposes a novel method for quantifying white blood cells in mallards using flow cytometry with the potential to implement this technique for samples collected in the field. The development of reagents, such as monoclonal antibodies that recognize leukocytes and thrombocytes originating from wild mallards represents a substantial amount of work and is a welcome addition to the understudied field of immunology of wild avian reservoirs. The gating strategy used to classify the cell populations using flow cytometry appears sound and reproducible and was evaluated relative to manual differential WBC counts using microscopy. The authors assess whether WBC numbers in mallards were effected by age, sex and infection with low pathogenic avian influenza. Age- and sex-specific differences in WBC numbers were detected which aligned with results of prior studies. Influenza infection did not correlate with any significant differences in WBC counts for which the authors suggest co-evolution between host and pathogen may be responsible. This is a timely paper, and the need for high throughput methods for quantifying blood cells that indicate health status of wild birds is compelling, especially against the backdrop of the current avian influenza virus outbreaks.

Response: Thank you very much for your detailed and constructive feedback on our study.

- 2. Reviewer's comment:** The authors might consider framing some hypotheses regarding WBC and the effects of age, sex and infection with influenza virus. This would be a useful addition to the Introduction. For instance, how would co-evolution impact host and viral factors? And is this similar to relationships observed for immune cells of other wild reservoirs such as bats and rats. There's little doubt that this is first and foremost a method development paper. However, the field testing of the method on mallards from Ottenby is an important aspect of the study which therefore should be framed with some predictions about the relationship between blood cells, demographic and infection ecology.

Response: Thank you for this input, we have now added some predictions and comparisons to other reservoir species in the introduction and discussion sections and highlighted how this may be affected by the co-evolutionary history of mallards with LPAIV.

Changes implemented: MS lines 74-76: "The ability of mallards to act as asymptomatic carriers of LPAI viruses, and to often survive HPAIV infections, is likely a result of a long co-evolutionary history of mallards with influenza viruses (7, 8)."

MS lines 108-113: “As sex and age affect the functions of the immune system (reviewed in 17, 18), we predicted that mallards of different age and sex may differ in their baseline WBC numbers. Other reservoir hosts, like bats, appear capable of avoiding immunopathology upon infection with certain pathogens with which they share an evolutionary history by restricting immune responses (19). We thus expected to see minor changes in WBC counts in LPAIV infected mallards.”

MS lines 492-493: “Likewise, evidence of lymphocytopenia has been observed during asymptomatic infection in the Marburg virus reservoir, the Egyptian rousette bat (46).”

- 3. Reviewer’s comment:** Would including Ct values from the influenza rRT-PCR provide useful context for understanding viral titer and perhaps where infected individuals are in the disease course? The lack of overall signal in WBC for infected juveniles might be confounded by the fact that early and late stages of infection are associated with different WBC profiles.

Response: We thank the reviewer for the excellent idea to include Ct values from the influenza RT-qPCR. We have now included some additional analysis in the manuscript investigating whether there is an association between the Ct value (as a proxy for viral titer) and the 1) number of WBCs and 2) body condition (body weight/wing length) in the mallards. Overall, none of the cell counts strongly correlated with the Ct value (neither when analysing females and males together, $p > 0.05$; Figure S6, nor separately, $p > 0.05$ data now shown). While there was a trend suggesting that AIV positive male juveniles with a lower body condition have higher viral titer, this association was not significant (Figure S8).

Interestingly, when looking at individual birds, the five birds with the lowest Ct value (highest viral titer) had relatively low CD8 T cell, B-cell and lymphocyte counts (Figure S6). This could potentially be an indication that individuals with a particularly high viral titer show stronger signs of lymphocytopenia, which we pointed out as one of the possible indicators of AIV infection in juvenile males in this manuscript.

Overall, at least in our study, Ct value was not a clear indicator for how strong immune response an individual bird show. Further, there was no clear pattern suggesting that birds with a low body condition have higher viral titer than birds with high body condition.

Changes implemented: MS lines 235-237: “We further investigated if there was an association between the RT-qPCR cycle threshold (Ct) value (a proxy for viral titer) and the WBC counts. None of the cell counts significantly correlated with the Ct value ($N = 46$, $p > 0.05$, Spearman’s $\rho = -0.22-0.11$; Supplementary Figure S6).”

MS lines 244-246: “Likewise, no significant correlation was detected between the Ct value and body condition of the birds (N = 46, $p > 0.05$, Spearman’s $\rho = 0.05-0.33$; Supplementary Figure S7).”

MS lines 395-396: “We used a Spearman rank correlation analysis to investigate if the viral RT-qPCR Ct value correlated with the WBC count of the individual birds.”

MS lines 517-525: “Cycle threshold (Ct) values obtained from RT-qPCR analyses indicate how much viral genetic material is in a sample, and can thus be used as a proxy for viral titer. Using the Ct value of all AIV positive juveniles, we investigated whether number of WBCs correlate with viral titer in mallards. Overall, none of the cell counts strongly correlated with the Ct value. In this study, Ct value was thus not a good indicator for the strength of immune response in individual birds. When looking at individual birds, the five birds with the lowest Ct value (indicating higher viral quantity) had relatively low CD8 T cell, B-cell and lymphocyte counts. This suggests that individuals with a particularly high viral titer may show stronger signs of lymphocytopenia. Additional studies are required to better understand the effect of viral titer on WBC counts in mallards.”

We further included two new figures in the supplementary files (Figure S6 and S8).

- 4. Reviewer’s comment:** Would also be useful to know if the IAV-positive mallards had sero-converted and therefore adaptive immunity had been mobilized. However, running serological assays to screen for antibodies is non-trivial and not vital to the outcome of the conclusions. I leave this at the discretion of the authors.

Response: We agree that serological information would have been a valuable addition to our study. Unfortunately, we do not have serological data for the birds included in this study. A previous study (Tolf et al 2012) suggests, however, that a large proportion of mallards at the Ottenby bird station are seropositive for AIV in late autumn.

Investigating a potential correlation of AIV antibody levels, infection status and changes in the WBC count in mallards would therefore be a highly interesting follow up study.

Tolf, Conny, et al. "Individual variation in influenza A virus infection histories and long-term immune responses in mallards." PLoS One 8.4 (2013): e61201.

- 5. Reviewer’s comment:** Including any additional data on host physiology, such as body condition, would be helpful to interpret the age, sex and infection status results. Studies of wild animals are complex due to the huge number of intrinsic and extrinsic factors that cannot be accounted for, unlike experimental studies. For example, the life-history and migratory behavior of each mallard cannot be known,

however, these are traits that may impact WBC as a measure of physiological health. Including this data or a statement on the ecological and physiological traits that were not measured in this study would be helpful for interpretation of the results.

Response: Thank you for this comment, it is true that there are many factors that may affect the blood cell counts that we could not account for in our study. We have included an additional statement raising this issue in the discussion section.

One host physiology parameter that we did record for all birds is body condition (body weight/wing length). We initially used this measure to detect individuals of low health, and accordingly excluded individuals with a body condition lower than three from our study. We have now included additional analyses to the manuscript investigating if body condition differed between mallards of different age, sex and infection status. Briefly, while there was an association between sex and body condition in juvenile mallards (Figure S7A), there was no evident difference between infected/non infected birds (Figure S7B).

Changes implemented: MS lines 239-247: “We investigated if body condition differed between different groups of mallards in our study. Juvenile females and males differed in their body condition, with females having lower body condition than males (ANOVA ($n=116$), $F(3, 88) = 7.99$, $p < 0.001$ and Tukey’s HSD post hoc test $t_{j-f,m} = 4.667$, $p_{j-f,m} < 0.001$; Supplementary Figure S7A). In contrast, there was no difference between infected and non-infected birds (two-sided t test; females ($n = 64$), $t(50.51) = 1.01$, $p = 0.319$ and males ($n = 67$), $t(46.38) = 0.63$, $p = 0.532$); Supplementary Figure S7B). Likewise, no significant correlation was detected between the Ct value and body condition of the birds ($N = 46$, $p > 0.05$, Spearman’s $\rho = 0.05-0.33$; Supplementary Figure S7). As no clear pattern was observed in body condition between infected and non-infected birds, this factor was not included in the main analysis.”

MS lines 398-402: “To investigate if body condition differed between mallards of different age and sex, we performed an ANOVA and a Tukey post-hoc test for pairwise means comparison. To test if body condition differed between infected and non-infected mallards, we used a two-sided t test. To test if the viral RT-qPCR Ct value correlated with the body condition of the individual birds we further used a Spearman rank correlation analysis.”

MS lines 455-464: “The high individual variation observed in the wild mallards was expected, as factors that may impact WBC counts (i.e. life-history and migratory behaviour) could not be determined for each bird. In fact, studies of wild animals are complex due to the large number of intrinsic and extrinsic factors that cannot be accounted for, unlike experimental studies. As sex and age affect the functions of the immune system (reviewed in 17, 18), we accounted for these factors in our study. To further account for individual variation in host physiology, we calculated the body

condition of each individual and compared it between mallards of different age, sex and infection status. Body condition differed between juvenile males and females, but not between AIV positive and negative birds. As body condition mainly correlated with sex, we did not include this factor in our analysis. Further studies are required to investigate the effect of intrinsic and extrinsic factors on WBC counts in mallards.“

We further included a new figure in the supplementary files (Figure S7).

May 25, 2023

Dr. Elinor Jax
Max-Planck-Institut für Verhaltensbiologie
Am Obstberg 1
Radolfzell
Germany

Re: Spectrum04351-22R1 (Evaluating effects of AIV infection status on ducks using a flow cytometry based differential blood count)

Dear Dr. Elinor Jax:

We appreciate your consideration of the reviewers' comments and for the submission of a revised version addressing all the issues raised.

I am pleased to inform you that your manuscript has been accepted, and I am forwarding it to the ASM Journals Department for publication. You will be notified when your proofs are ready to be viewed.

Sincerely,

Rafael A. Medina
Editor, Microbiology Spectrum
